# Automated quantitative histology reveals vascular morphodynamics during Arabidopsis hypocotyl secondary growth

**Martial Sankar[1†], Kaisa Nieminen[1†], Laura Ragni[1†], Ioannis Xenarios[2], Christian S Hardtke[1*]**

[1]Department of Plant Molecular Biology, University of Lausanne, Lausanne, Switzerland; [2]Vital-IT, Swiss Institute of Bioinformatics, Lausanne, Switzerland

**Abstract** Among various advantages, their small size makes model organisms preferred subjects of investigation. Yet, even in model systems detailed analysis of numerous developmental processes at cellular level is severely hampered by their scale. For instance, secondary growth of Arabidopsis hypocotyls creates a radial pattern of highly specialized tissues that comprises several thousand cells starting from a few dozen. This dynamic process is difficult to follow because of its scale and because it can only be investigated invasively, precluding comprehensive understanding of the cell proliferation, differentiation, and patterning events involved. To overcome such limitation, we established an automated quantitative histology approach. We acquired hypocotyl cross-sections from tiled high-resolution images and extracted their information content using custom high-throughput image processing and segmentation. Coupled with automated cell type recognition through machine learning, we could establish a cellular resolution atlas that reveals vascular morphodynamics during secondary growth, for example equidistant phloem pole formation.

***For correspondence:** christian.hardtke@unil.ch

[†]These authors contributed equally to this work

**Reviewing editor**: Jan Traas, Ecole normale supérieure de Lyon, France

## Introduction

Model organisms have proven essential for dissecting the molecular-genetic control of biological processes in both animals and plants (*Meyerowitz, 2002*; *Brenner, 2009*). Typically, they have been chosen according to a number of criteria, including a small, diploid genome, a short generation time, and easy lab culture. Another frequent feature is their small size, which allows cultivation of numerous individuals to enable large-scale genetic analyses as well as easy observation of developmental processes by microscopy. Fulfilling all these criteria, *Arabidopsis thaliana* (Arabidopsis), a small, annual dicotyledon of the *Brassicaceae* family, is the model of choice for developmental biology of higher plants (*Meyerowitz, 1989*). Various central processes of the plant life cycle, for example embryogenesis, root meristem organization or flower development can be examined at high spatio-temporal resolution in Arabidopsis. Moreover, in many instances live imaging at (sub-) cellular level is possible through microscopy techniques, including confocal microscopy, which is aided by the transparency of whole organs, such as the root, or at least the outermost tissue layers. However, such investigation is limited by organ depth, which can increase dramatically with organ size. For example, while the meristematic and differentiation regions of the root tip comprise a mere 5–6 dozen cells in the radial dimension and can be imaged all across using state-of-the-art microscopes, cell number rapidly increases proximal, towards the mature root (*Dolan et al., 1993*). At the same time, the organization of the root tissue layers rearranges from a partially radial, partially bilateral symmetry towards full radial symmetry, concomitant with the formation of cylindrical secondary meristems and the replacement of the outer cell layers by a new protective outside tissue. Thus, eventually the mature root acquires the same overall organization as the mature aboveground stems, that is a few cell layers of protective tissue produced by an underlying cork cambium that surround the vascular tissues. The latter are produced by another

**eLife digest** Our understanding of the living world has been advanced greatly by studies of 'model organisms', such as mice, zebrafish, and fruit flies. Studying these creatures has been crucial to uncovering the genes that control how our bodies develop and grow, and also to discover the genetic basis of diseases such as cancer.

Thale cress—or *Arabidopsis thaliana* to give its formal name—is the model organism of choice for many plant biologists. This tiny weed has been widely studied because it can complete its lifecycle, from seed to seed, in about 6 weeks, and because its relatively small genome simplifies the search for genes that control specific traits. However, as with other much-studied model systems, understanding the changes that underpin the development of some of the more complex tissues in *Arabidopsis* has been severely hampered by the shear number of cells involved.

After it has emerged from the seed, the plant's first stem will develop from a few dozen cells in width to several thousand cells with highly specialized tissues arranged in a complex pattern of concentric circles. Although this stem thickening process represents a major developmental change in many plants—from *Arabidopsis* to oak trees—it has been under-researched. This is partly because it involves so many different cells, and also because it can only be observed in thin sections cut out of the plant's stem.

Now Sankar, Nieminen, Ragni et al. have developed a novel approach, termed 'automated quantitative histology', to overcome these problems. This strategy involves 'teaching' a computer to automatically recognize different plant cells and to measure their important features in high-resolution images of tissue sections. The resulting 'map' of the developing stem—which required over 800 hr of computing time to complete—reveals the changes to cells and tissues as they develop that allow the transport of water, sugars and nutrients between the above- and below-ground organs. Sankar, Nieminen, Ragni et al. suggest that their novel approach could, in the future, also be applied to study the development of other tissues and organisms, including animals.

cylindrical secondary meristem, the vascular cambium, which produces xylem tissues towards the inside and phloem tissues towards the outside (*Nieminen et al., 2004*; *Groover and Robischon, 2006*). The activity of the cambial stem cells drives the radial expansion of roots and stems, a process termed 'secondary growth'.

Formation of xylem tissues through secondary growth is the main process of durable biomass accumulation in plants and most prominent in tree trunks (*Groover and Robischon, 2006*; *Spicer and Groover, 2010*). In Arabidopsis, substantial secondary growth is not only observed at later stages of root development, but also in the hypocotyl, the embryonic stem (*Chaffey et al., 2002*; *Sibout et al., 2008*) (*Figure 1A*). Consistent with the hypocotyl's role as critical junction between the root and shoot systems that limits the reciprocal transfer of edaphic resources and photosynthetic metabolites, its secondary growth occurs throughout most of the Arabidopsis life cycle and in some ways resembles the radial expansion of tree trunks. The hypocotyl initiates secondary growth shortly after seedling establishment, once its cell elongation growth along the main body axis has seized (*Sibout et al., 2008*; *Ragni et al., 2011*). Thus, unlike in post-embryonic stems, secondary growth in the hypocotyl is not obscured by parallel elongation growth, making it an ideal model system for this process.

Previous work has identified two principal phases of hypocotyl secondary growth, an early phase of proportional growth, when the cambium produces phloem and xylem tissues at roughly equal rates, and a later phase of xylem expansion, when the relative production of xylem dominates the radial expansion (*Chaffey et al., 2002*; *Sibout et al., 2008*). Early phase xylem consists mainly of the interconnected xylem vessels (terminally differentiated, dead cells with perforated, thick cell walls that are the actual conducts for water and solutes) and xylem parenchyma cells. Early phase phloem comprises the sieve elements (interconnected, enucleated but alive cells that perform the actual transport of the phloem sap), companion cells (which provide basic metabolism for sieve elements and are responsible for loading and un-loading of phloem sap cargo) and phloem parenchyma cells. In the xylem expansion phase, both xylem and phloem also start to differentiate fibers (cells with thick secondary cell walls that provide structural support), which can be formed from parenchymatic precursor cells.

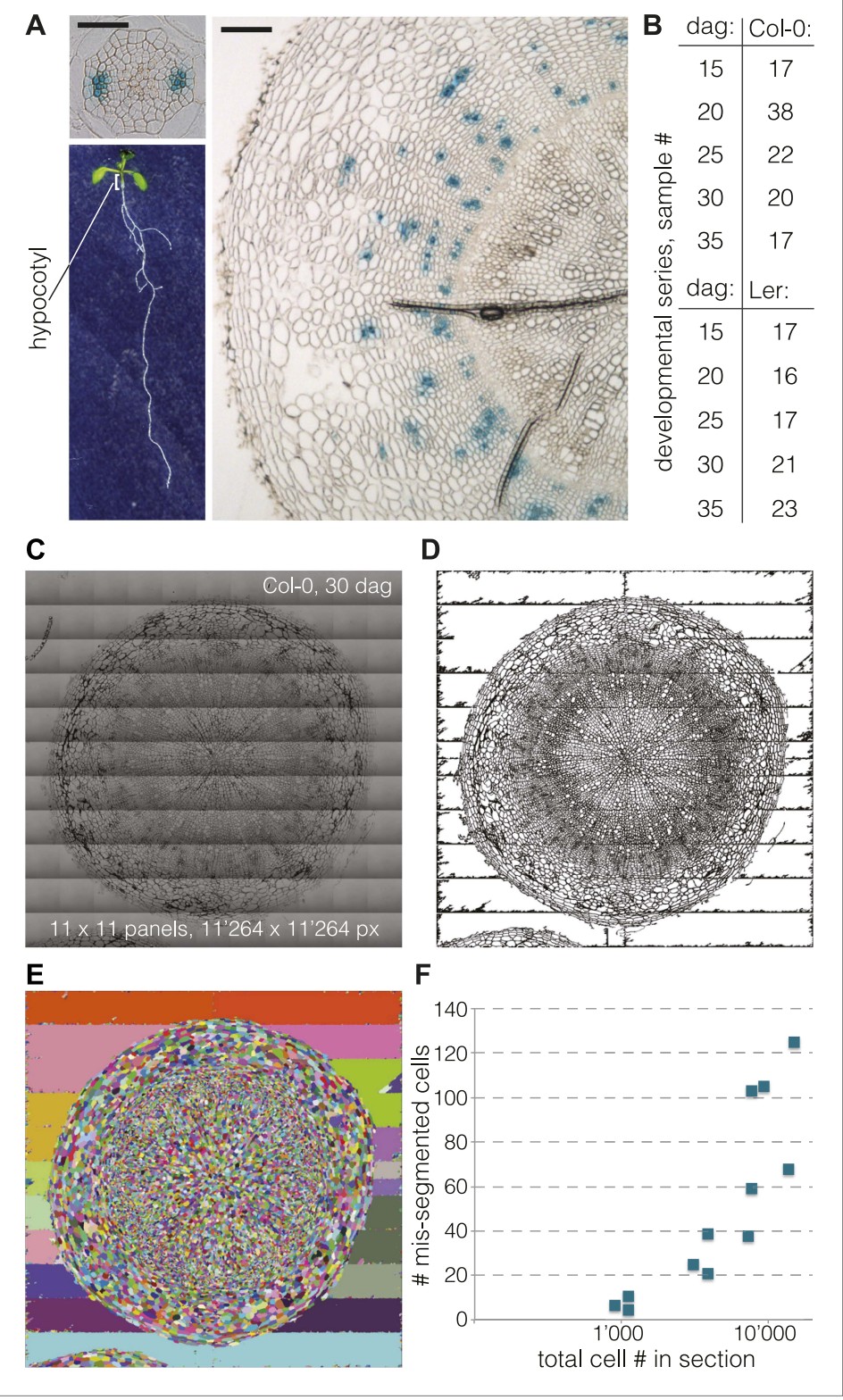

Figure 1. Cellular level analysis of Arabidopsis hypocotyl secondary growth. (A) Light microscopy of cross sections obtained from Arabidopsis hypocotyls (organ position illustrated for a 9-day-old seedling, lower left) at 9 dag (upper left) and 35 dag (right). Size bars are 100 µm. Blue GUS staining due to the presence of an *APL::GUS* reporter gene in this Col-0 background line marks phloem bundles. (B) Overview of the developmental series

*Figure 1. Continued on next page*

*Figure 1. Continued*

(time points and distinct samples per genotype) analyzed in this study. (**C**) Example of a high-resolution hypocotyl section image assembled from 11 × 11 tiles. (**D**) The same image after pre-processing and binarization, and (**E**) subsequent segmentation using a watershed algorithm. (**F**) Number of mis-segmented cells as determined by careful visual inspection in 12 sections, plotted against the total number of cells per section (log scale).

It has been shown that the transition between the early phase and the xylem expansion phase is triggered by the onset of flowering (*Sibout et al., 2008*), through a mobile shoot-derived signal, the plant hormone gibberellin (*Ragni et al., 2011*). In these studies, the transition had been defined as a shift in the relative occupancy of overall xylem vs overall phloem tissue in hypocotyl cross sections. However, as only the overall areas of combined xylem and phloem tissues were considered, it remains unclear what the transition represents at the cellular level. Various scenarios could be envisioned, for instance the relative expansion of xylem might be a consequence of increased post-cambial proliferation during xylem differentiation, or of increased cambial stem cell activity toward the xylem side, or the inverse with respect to phloem. To distinguish between these possibilities proved to be very difficult due to the absence of information about the cellular dynamics during the secondary growth process. Moreover, such investigation is severely hampered by the fact that this process is not amenable to live imaging and can only be monitored invasively, through histological cross sections, thereby killing the individual sample under investigation. Thus, a quantitative understanding of the temporal progression of secondary growth can only be acquired by a high-throughput approach that monitors enough cross-sections from distinct hypocotyls of the same age to provide statistically solid data. In conjunction with the large number and morphological diversity of the cells that constitute this tissue, a quantitative understanding of the cell proliferation, differentiation, and patterning events by conventional means, that is simple visual inspection of cross sections, is out of reach. Therefore, we established an automated histology approach to create a cellular resolution atlas that reveals the vascular morphodynamics during hypocotyl secondary growth. Our data reveal substantially different secondary growth dynamics in two genotypes as well as emerging patterns of cell orientation over time and a constantly equidistant production of phloem poles by the cambium.

## Results

The goal of our study was to develop a universally applicable, 'automated quantitative histology' approach that could be applied to provide a comprehensive, quantitative analysis of the vascular morphodynamics during hypocotyl secondary growth in Arabidopsis. For analysis we chose two common laboratory accessions, Columbia-0 (Col-0) and Landsberg *erecta* (Ler), which have already been shown to display divergent secondary growth dynamics (*Ragni et al., 2011*).

### Raw data collection and rough analysis

Based on the secondary growth progression observed during pilot experiments, we chose to analyze five time points in detail, starting at 15 days after germination (dag), when a full cambium is established and the initial outer epidermal and cortex cell layers are already or about to be shed. This was followed by additional sampling at 20, 25, 30 and up to 35 dag, when the plants had seized formation of new flowers (*Figure 1B*). Plants were grown in soil in a 16 hr light–8 hr dark cycle at 22°C with 150 µE light intensity. To minimize variation due to environmental conditions and between experiments, all plants were grown in parallel in a randomized design. In our conditions, all plants of both genotypes flowered at 17 dag ±1 d. For each time point, 50 seedlings were initially planted with the goal to eventually harvest 40 hypocotyls, which were fixed and embedded for sectioning. Embedding was performed using plastic resin, which proved to be the only robust method to acquire 3 µm thin cross-sections while conserving the cellular structure. A first observation by light microscopy after toluidine blue staining confirmed the integrity of the samples and allowed a first rough analysis of secondary growth progression based on overall transverse area (excluding any remaining epidermal or cortex layers) and the proportion occupied by the xylem. Whereas the average hypocotyl stele diameter was ca. 0.3 mm in Col-0 and ca. 0.15 mm in Ler at 15 dag, the radial expansion resulted in an average diameter of 1.6 mm in Col-0 and 1.1 mm in Ler at 35 dag. Concomitantly, relative xylem area increased from 12% to 29% in Col-0, and from 31% to 47% in Ler, confirming previous observations (*Ragni et al., 2011*).

## Acquisition and segmentation of high-resolution images

To obtain accurate quantitative parameters of secondary growth progression, we implemented a segmentation procedure to extract the cellular features from the cross sections. To allow reliable identification of small cells, such as cambial cells, with standard segmentation software, we obtained images of the cross sections with a light microscope at 40 X magnification. Our strategy was to produce ultra-high resolution images of 1024 × 1024 pixels, which would allow a very fine discrimination of individual cell boundaries, the critical requirement for the subsequent image segmentation process. Because the resolution was too high to fit any single cross section into a single image, we used the tiling function of the microscope to fuse 1024 × 1024 pixel subpanels into single images for each cross section. Individual cross section images subjected to segmentation were thus assembled from a minimum of 9 (3 × 3) up to 144 (12 × 12) panels (*Figure 1C*). This procedure permitted information extraction from the whole section without inference or data loss. To this end, we developed a custom, fully automated image processing and segmentation pipeline. This pipeline pre-processes the images (gamma correction, contrast and brightness adjustment) and discards noise pixels after binarization (*Figure 1D*) before segmentation using a watershed algorithm (*Figure 1E*). The pipeline is fully automated and robust and typically performed at more than 99% accuracy (i.e., less than 1% of mis-segmented cells) across the scale of images (*Figure 1F*). However, because CPU time scaled exponentially with image size, taking ca. 8 min. for a 15 dag sample but ca. 1000 min for a 35 dag sample, computation eventually became limiting for our endeavor. Thus, we restricted our analysis to ca. 20 selected cross sections per genotype and time point (i.e., 208 cross sections in total, requiring ca. 800 hr of total CPU time) (*Figure 1B*), which gave statistically robust quantitative data.

## Computation of cellular descriptors

Overall median cell number at 15 dag was 883 for Col-0 and 260 for Ler. At 35 dag, it had increased to 18'124 and 11'026, respectively, indicating higher overall secondary growth in Col-0, but higher relative secondary growth in Ler (i.e., a ca. 42-fold vs a ca. 21-fold increase in cell number). Together with the overall increase in total transverse area (from ca. 70'000 $\mu m^2$ at 15 dag to ca. 2 million $\mu m^2$ at 35 dag and ca. 11'000 $\mu m^2$ to ca. 1 million $\mu m^2$ for Col-0 and Ler, respectively), this suggests significantly different secondary growth dynamics in the two genotypes. However, these overall averages can be misleading because of the already observed differences in relative tissue abundance. Thus, we advanced towards our goal of a full cellular resolution analysis by computing 16 cellular descriptors that represent the geometric characteristics of cell shape and relative cell position (*Supplementary file 1A*). The initial set of descriptors was extracted from the segmented images using the EBImage R package (*Pau et al., 2010*). This toolbox computes morphological features by calculating the 2nd-order covariance matrix of the image moments, which is equivalent to fitting an ellipsoid to an object. From these data, we computed additional features, including the position of the cells given by their polar coordinates and the cell incline angle (see below), thereby taking full advantage of the cylindrical morphology of the hypocotyl cross sections.

## Supervised learning for automated cell type recognition

Although the descriptors provided an overview of the cell sizes, shapes and positions within the sections, they did not provide a straightforward indication of the tissue that individual cells belong to. To overcome this limitation, we sought to develop automated cell type recognition that uses the descriptors as an input for cell type classification. To this end, we performed a supervised classification using the support vector machine algorithm (SVM) (*Cortes and Vapnik, 1995*). Briefly, the SVM classifier principle is to find the optimal decision boundary between classes by maximizing the margin hyperplanes (the geometrical representation of the decision boundaries in multi-dimension) between the support vectors. The training set was a subset of our data that comprised a total of 3'144 manually labeled cells, dispatched into two sections per time point and genotype (*Supplementary file 1B*). This set was split into a learning set comprising two-thirds of the data, and a test set constituted from the remaining cells. The former subset was used to build the classifier whereas the latter was employed for validation. The performance was assessed using the V-fold cross validation method, which consists of five randomly permutated reiterations of training and test sets to maximize the test set prediction error rate. Feature selection is a well-known pivotal issue in machine learning, and indeed the best combination of descriptors was critical in automated cell type classification and varied with the time point and genotype analyzed, mainly because cell type-specific position can vary with the age of the section.

Thus, we developed a greedy algorithm for feature selection based on the 16 initial descriptors. This allowed us to select descriptors according to their importance in classifier performance (*Figure 2—figure supplement 1*), such that we could build one optimized classifier with respect to a given time point. In general, we selected the combination with the least number of descriptors, the lowest variation and the highest cross-validation performance with respect to the training/test set permutations. Finally, another key criterion in classifier selection was to minimize performance trade-off across different cell types, that is classifiers that scored high in correct recognition of all the different cell types (the selected classifiers are described in *Supplementary file 1C*). Across all sections and time points, a common set of five distinct cell type categories (*Supplementary file 1D*) could be classified and quantified, that is (i) xylem vessels and parenchymatic cells, (ii) xylem fibers, (iii) cambial cells, (iv) phloem bundle cells (companion cells and sieve elements) and (v) parenchymatic phloem cells (including any of the rare phloem fibers) (*Supplementary file 1E*). Although more categories (e.g., xylem vessels and parenchyma cells separately) could be reliably distinguished at individual time points using other classifiers, we restricted our analyses to these five for the sake of a coherent temporal description of secondary growth progression. For these categories, our purely morphology- and position-based approach identified cells with an average accuracy of 88% and a median accuracy of 95% across the n = 50 cell type category X time point X genotype matrix.

## Automated quality control and refinement of cell type recognition

Whereas the automated recognition with these classifiers was thus sufficiently accurate for most cell type categories to extract quantitative data about secondary growth progression from the typically thousands of cells per section, the recognition of the xylem vessel and parenchyma cells behaved as an outlier, with lower accuracy especially at later stages. We also noticed that xylem cell types were frequently assigned to cells outside of the xylem area's average radius. This was particularly prevalent at the later stages of development and could be pinpointed to a frequent confusion between xylem vessels and phloem parenchyma cells, which increasingly resemble each other in their outlines as secondary growth proceeds. However, discarding the problematic sections based on stringent criteria would have meant the exclusion of 33% and 40% of sections for Col-0 and Ler, respectively. To tackle these problems, we developed an automated pipeline for quality control. This procedure was based on manually created mask images that specified both the xylem area and the whole section area of the 208 samples (*Supplementary files 2 and 3*). Segmentation of the mask images allowed us to filter out noisy objects outside the sections' average radius distance, mostly mis-segmented objects that either represented dirt contaminations or shed epidermal or cortex cells. The tool also automatically corrected the mis-assigned xylem cell identities by taking advantage of the mask-defined xylem area of the quality control filter. This correction refined the cell type recognition results and permitted all sections to pass the filtering step. An overview of the entire computational pipeline is shown in *Figure 2A*.

## Overall similar but temporally shifted vascular patterning in the two genotypes

For a first overview of secondary growth progression, we used the thus extracted cellular data to define phenotypic profiles (phenoprints) for each time point and genotype, comprised of the global (e.g., cross-section size or total cell count) and cell type-specific (e.g., relative proportion of a particular cell type category or feature distribution) statistics (*Figure 2B*) (the feature description data for all cells of all sections is provided in data files 1 and 2, the corresponding normalized data used for machine learning and the determined cell type identities are provided in the data files 3 and 4, all available in the Dryad data repository under doi: 10.5061/dryad.b835k (*Sankar et al., 2014*)). The phenoprints consisted of a set of eight multi-parametric descriptors, which was informative for the normalized values (*Supplementary file 4*) that were used to perform a principal component analysis (*Figure 3A*). The computed correlation matrix was projected into a two-dimensional coordinate system, with the first two principal components explaining 76% of the variation. The first component opposed the larger phenoprint stages (30–35 dag in both genotypes) with the smallest (Ler 15d), with proportionally less cambium in the older stages. The second component associated variables of large phloem proportion and inexistent or low fiber content (Col-0 15 dag, Ler 25 dag, Col-0 20 dag, Col-0 25 dag). The analysis also revealed larger angle spans for Ler as compared to Col-0 above all between 15 dag and 25 dag, suggesting substantial morphological changes during the early stages. At later time points, the two genotypes increasingly clustered together, indicating an initially slower development

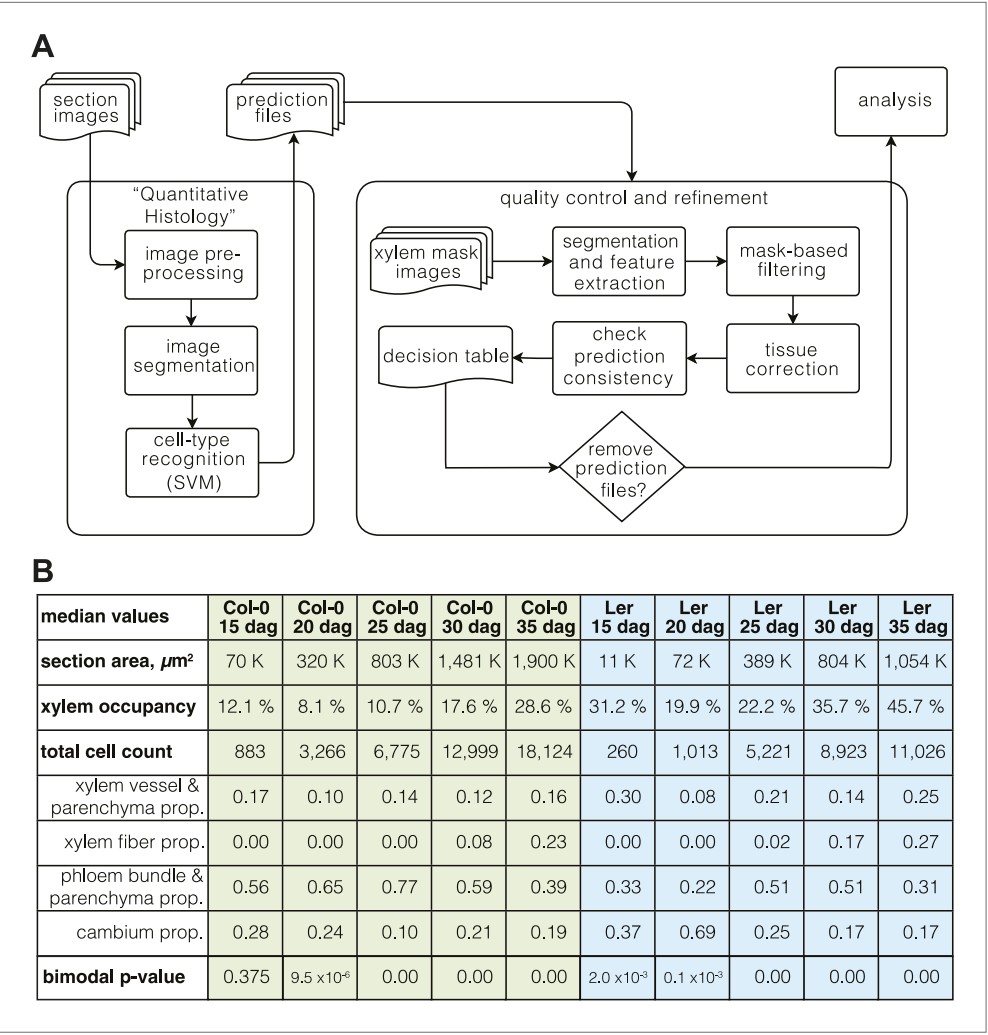

**Figure 2**. The 'Quantitative Histology' approach. (**A**) Overview of the computational pipeline from image acquisition to analysis. (**B**) 'Phenoprints' for the different genotypes and developmental stages.

The following figure supplements are available for figure 2:

**Figure supplement 1**. An example of classifier selection through V-fold cross validation.

in Ler that however eventually caught up with Col-0. Overall, the phenoprint clustering suggests a conserved sequence of development from one distinct morphological pattern to another, albeit with a different temporal progression in Col-0 vs Ler.

## Reduced phloem cell proliferation is the cause of higher xylem area occupancy in Ler

Previous studies (*Ragni et al., 2011*) have shown that Ler has a higher ratio of xylem area to phloem area than most other accessions, including Col-0. Our quantification also confirmed that overall radial expansion of Ler was reduced as compared to Col-0 (*Figure 3B*). However, xylem area expansion rate was nearly equal in both genotypes, which combined with lower overall radial expansion necessarily resulted in higher xylem occupancy in Ler (*Figure 3C*). In the temporal trend, two distinct phases of xylem occupancy could be distinguished. Initially, it decreased or remained stable between 15 and 25 dag, followed by an increase between 25 and 35 dag. Whereas these tendencies were similar in both genotypes up to 30 dag, Ler differed in that its xylem area increased steadily, eventually occupying almost 50% of the total transverse area at 35 dag. Quantification of cell proliferation confirmed that the number of xylem cells and the xylem cell proliferation rate were close in both genotypes (*Figure 3D*), however the total number of cells

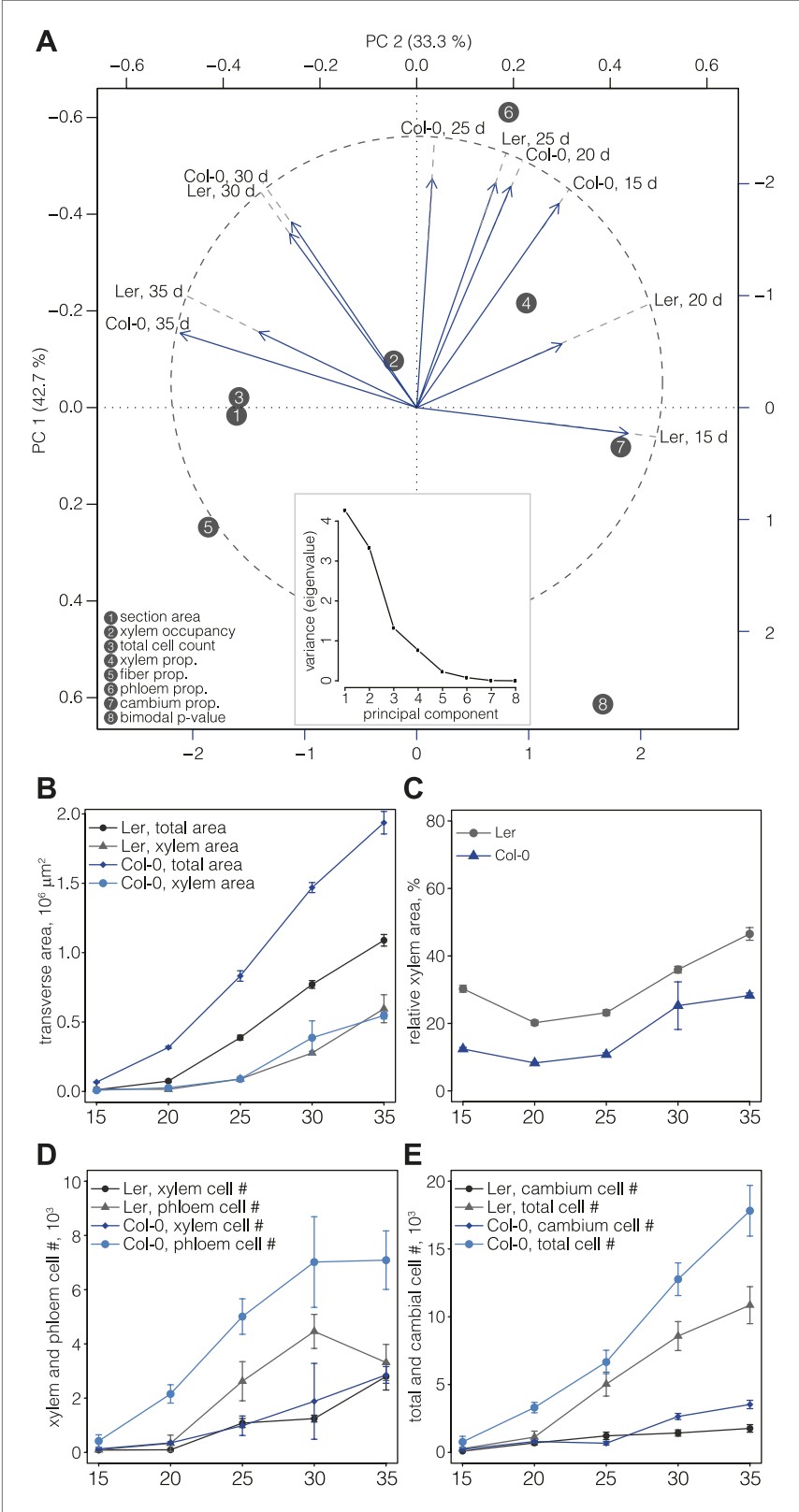

**Figure 3**. Progression of tissue proliferation. (**A**) Principal component analysis (PCA) of the phenoprints shown in *Figure 2B*, performed with normalized values (*Supplementary file 4*). The inlay screeplot displays the proportion of total variation explained by each principal component. (**B–E**) Comparative plots of parameter progression in the *Figure 3. Continued on next page*

*Figure 3. Continued*

two genotypes. In (**D**), xylem represents combined vessel, parenchyma, and fiber cells, phloem represents combined phloem parenchyma and bundle cells. Error bars indicate standard error.

in Ler was ca. twofold lower than in Col-0 (*Figure 3E*). Moreover, the phloem proliferation rate was more than twofold lower in Ler, with stagnation in phloem cell number between 30 and 35 dag (*Figure 3D*) explaining the high xylem tissue occupancy at 35 dag. The increase of cambium cell number in Col-0 as compared to Ler at later stages of development (*Figure 3E*) likely contributed to this difference. In summary, our results suggest that a plateau in cambial growth combined with stagnating phloem proliferation is responsible for overall reduced radial growth but relatively increased xylem expansion in Ler.

## Visualization of vascular morphodynamics through combined plots of cell size and incline angle

Of the descriptors extracted by our computational approach, the incline angle proved to be most useful in detecting and illustrating the substantial features of vascular organization during secondary growth progression. The incline angle represents the deviation of the major axis of a cell with respect to the radius emanating from the manually defined center point of the cross section (*Figure 4—figure supplement 1*). We calculated the incline angle $\theta$ (in radians) as follows:

$$\theta = \left| \arccos\left( \frac{x.r}{\|x\|.\|r\|} \right) - \pi/2 \right|$$

where x and r are vectors, corresponding to the major axis of the cell and the radius running from the cell center, respectively. A value of zero represents perfectly orthoradial (i.e., tangential or periclinal) orientation of the major axis, and a value of π/2 represents perfectly radial (i.e., anticlinal) orientation. Plotting the incline in combination with cell size created informative simplified visualizations of our cross sections (*Figure 4A–B*). In these, concentric areas of cell orientation are evident, with a central area of mainly large and radially oriented (high incline) cells, representing the xylem cell categories. This area is surrounded by the cambium, depicted as a ring of small and orthoradially oriented (low incline) cells and, reaching the periphery, a zone comprising a bulk of mainly larger, orthoradially oriented cells representing the phloem area. Following the plots across the time points allowed us to reveal the vascular morphodynamics as a function of incline.

## Local variation and rearrangement of incline angles support distinct phases of vascular patterning

Interestingly, the spatio-temporal dynamics of the overall incline (i.e., covering the whole section cell content at a given time point) captured the distinct phases of secondary growth progression described above. This could be visualized in violin plots (*Figure 4C–D*), where the incline angle was uniformly distributed at 15 dag in Col-0 (Hartigans' dip test p>10$^{-3}$), meaning that no distinguishable vascular organization of cell orientation was yet built up. Starting at 20 dag, a first peak towards lower values of incline emerged and persisted until 35 dag. At 30 dag, a second peak towards higher values of incline arose, giving shape to a discernable bimodal distribution (Hartigans' dip test p<2.2 × 10$^{-6}$) (*Figure 4C*). In Ler, the pattern was different in that a broad, slightly skewed distribution with a median value towards the lowest values of incline was observed at 15 dag, followed by a broad, slightly bimodal distribution at 20 dag (Hartigans' dip test p<10$^{-4}$) (*Figure 4D*). At the later time points, sharp bimodal-shaped density curves supported the coexistence of two populations of cells, a mostly radially and a mostly orthoradially oriented one (Hartigans' dip test p<2.2 × 10$^{-6}$), similar to Col-0.

## Spatio-temporal patterning of inclines

Plotting the incline of individual cells according to their radial position (i.e., distance from a cross section's center) and over time points, we could follow the rearrangement in more detail. Normalization allowed us to pool the cells from all sections from a given time point and perform relative comparisons between them. Fitting these cloud distributions with locally weighted linear regression (i.e., lowess) revealed the essential data trends (*Figure 5*). In Col-0, the spatial distribution of the cell incline displayed unexpected temporal dynamics. At 15 dag, a wavy line described the point cloud, meaning

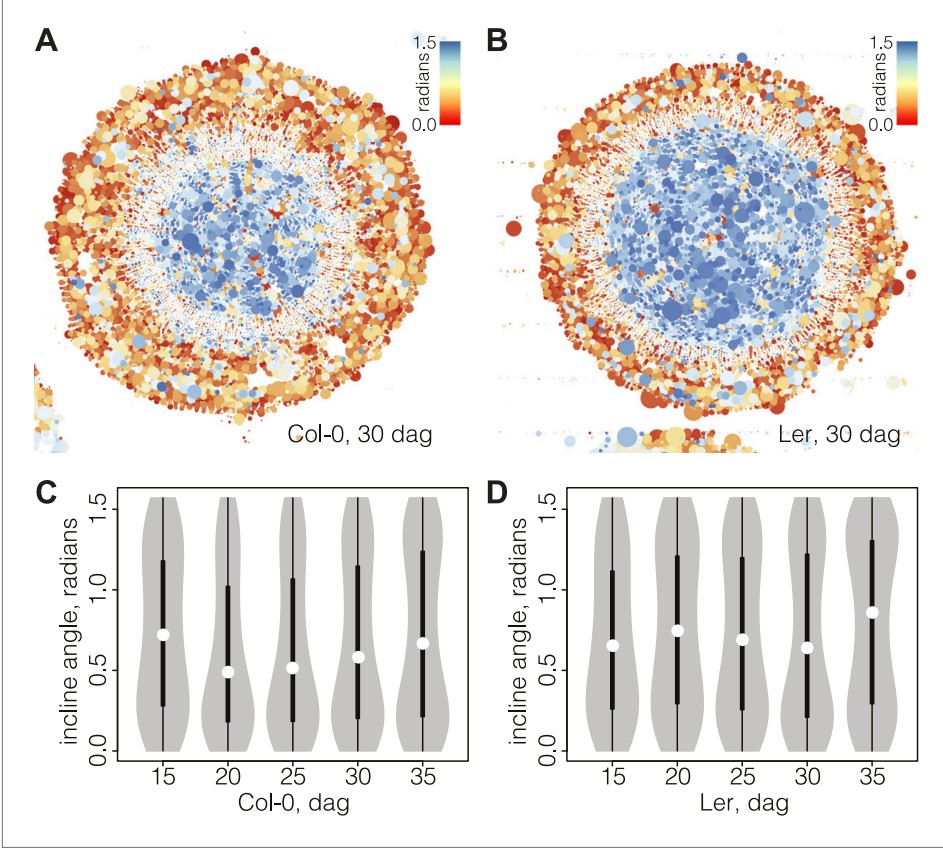

**Figure 4**. Bimodal distribution of incline angle according to position. (**A** and **B**) Spatial distribution of cell incline angle illustrates the vascular organization in Ler (**B**) as compared to Col-0 (**A**) at later stages of development, for example 30 dag. The size of the disc increases with the area of the cell. Blue color indicates radial cell orientation, red orthoradial. (**C** and **D**) Violin plots of incline angle distribution, illustrating increasingly bimodal distribution coincident with refined vascular organization and different dynamics of the process in the two genotypes.

The following figure supplements are available for figure 4:

**Figure supplement 1**. An illustration of the incline angle.

---

that a radial vs orthoradial tissue boundary was not yet distinguishable (*Figure 5A*). However, around 20 and 25 dag, vascular organization emerged as a plateau of largely radial orientation close to the center that corresponded to xylem cells, followed by a steep decrease to lower incline values in the cambium and phloem tissues (*Figure 5C,E*). Once this pattern was established, the plateau of xylem enlarged while the span of the orthoradial cell layers narrowed, concomitant with the occurrence of xylem fibers and expansion of the xylem area (*Figure 5G,I*). We also observed a decrease in the variation spread of incline in cambial cells over time. This reflected the progressive enlargement and organization of the cambium, which appeared to be completed as late as 30 dag, confirming continuous refinement of vascular patterning during secondary growth. A largely similar pattern of events was observed in Ler (*Figure 5B,D,F,H,J*), however, the final organization appeared more bimodal than in Col-0, which might reflect the above described decline of relative phloem area size.

## Cell proliferation and division plane switching is largely restricted to the cambium

The distribution of inclines also had possible implications for the orientation of cell divisions, in the sense that mostly radial orientation could be an indicator for the prevalence of anticlinal division planes, whereas mostly orthoradial orientations could be an indicator for the prevalence of periclinal ones. Visual inspection of cross sections suggested that this is not the case however, also revealing a remarkable rarity of post-cambial cell divisions. In the xylem area, practically no post-cambial divisions

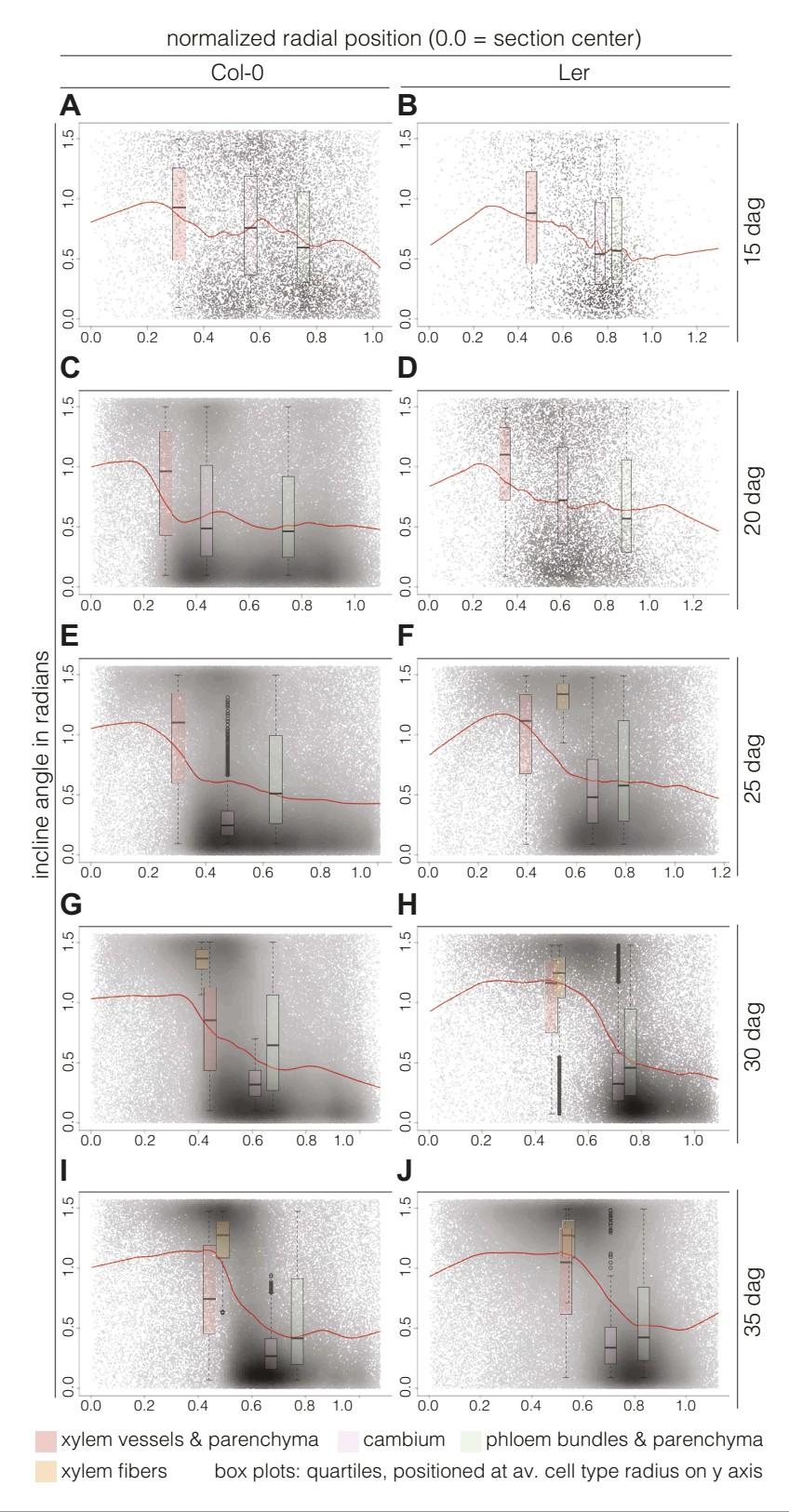

**Figure 5**. Distinct local organization of incline angle during hypocotyl secondary growth progression. (**A–J**) Density plots of cell incline angle vs radial position for the two genotypes at the indicated developmental stages, representing all cells across all sections for a given time point. The red lines represent the fit of these cloud distributions
*Figure 5. Continued on next page*

*Figure 5. Continued*

with locally weighted linear regression (i.e., lowess), revealing the essential data trends. All sections were normalized from 0.0 (the manually defined center) to 1.0 (the average radius in a set of sections as determined by the average distance of the outermost cells from the center for individual sections). Box plots indicate the quartiles of the radian distribution for each cell-type class and are placed at the average position of the cell type with respect to the y axis. Outliers are shown as circles.

The following figure supplements are available for figure 5:

**Figure supplement 1**. Analysis of cell number in defined xylem regions of different size.

were observed (*Figure 5—figure supplement 1*) and radial cell files were generally continuous with the adjacent cambial files. Following such cell files also suggested that cellular growth led to a switch in xylem cell incline angle orientation. Whereas xylem cells that emerged from the cambium still retained the orthoradial orientation, cellular growth eventually resulted in a switch towards a radial orientation. Such switching was not observed in the phloem, consistent with the prevalence of orthoradial inclines. Similar to the xylem however, phloem cells were typically in continuity with the corresponding cambial cell files, and practically no cell divisions, neither anticlinal nor periclinal, were observed. Importantly however, this was only observed for files of phloem parenchyma cells. The exceptions to this were cell files that ended up in vascular bundles. In these, numerous post-cambial divisions could be observed, both in the anticlinal and periclinal orientations. Finally, as expected the vast majority of cell divisions was observed in the cambium. Mostly, they occurred in a perfect periclinal orientation, but we also observed numerous interspersed anticlinal divisions that are necessary to keep up with overall radial expansion. In summary, the radial expansion of hypocotyls appeared to be mostly driven by cambial activity and very little by post-cambial cell divisions.

## Phloem pole formation displays a precise periodicity

Since there appeared to be no cessation of cell division in the cell files connecting the cambium and the vascular bundles, the data suggested that the patterning of phloem pole position might already be laid down in the cambium. Although such patterning was not evident from visual inspection of phloem pole distribution, a density map representation of phloem bundle cells suggested a spatial pattern of phloem poles positioning around the central xylem (*Figure 6A*). These density maps typically had limited resolution power around the cambial area, since newly born poles contain fewer bundle cells but are close in space, leading to a high and broad intensity region. For a more precise mapping of phloem pole positions, we thus analyzed 20, 25, and 30 dag sections obtained from transgenic Col-0 plants that expressed a beta-glucuronidase (GUS) reporter gene under the control of the phloem bundle-specific *ALTERED PHLOEM DEVELOPMENT (APL)* gene promoter (*Bonke et al., 2003*) (*Figure 1A*). Along a concentric ring-shaped region of interest across the emerging phloem poles, the latter appeared as dark foci of GUS staining with higher pixel intensity. In image analyses, these were detectable as intensity spikes (after noise reduction through the application of Gaussian blur, mainly to dampen background originating from the opacity of cell walls) (*Figure 6B*). Statistical analysis of the position of emerging phloem poles around the cambium revealed their spacing with a constant arc interspace distance. That is, the distance between emerging phloem poles remains constant over time as the cambial circumference enlarges. This was revealed by determination of the corresponding probability density function for the distance between the spikes by an automated Bayesian model (*Granqvist et al., 2012*), which indicates a constant arc interspace distance (*Figure 6C*) with a span of ca. 140 μm, suggesting that vascular bundle formation is a patterned rather than a stochastic process.

## Discussion

Secondary growth is a major developmental process in dicotyledons, including herbeaceous plants such as Arabidopsis (*Chaffey et al., 2002*; *Sibout et al., 2008*; *Elo et al., 2009*); however, it is comparatively an under-researched trait. In part, this can be attributed to the difficulty of investigating secondary growth in situ in a non-invasive manner, in part to the relatively big scale of the process. Both complications also contribute to the fact that a comprehensive description of secondary growth dynamics at the cellular level is still missing. In this study, we aimed to provide a robust quantitative description of secondary growth that could serve as a reference frame for future investigations.

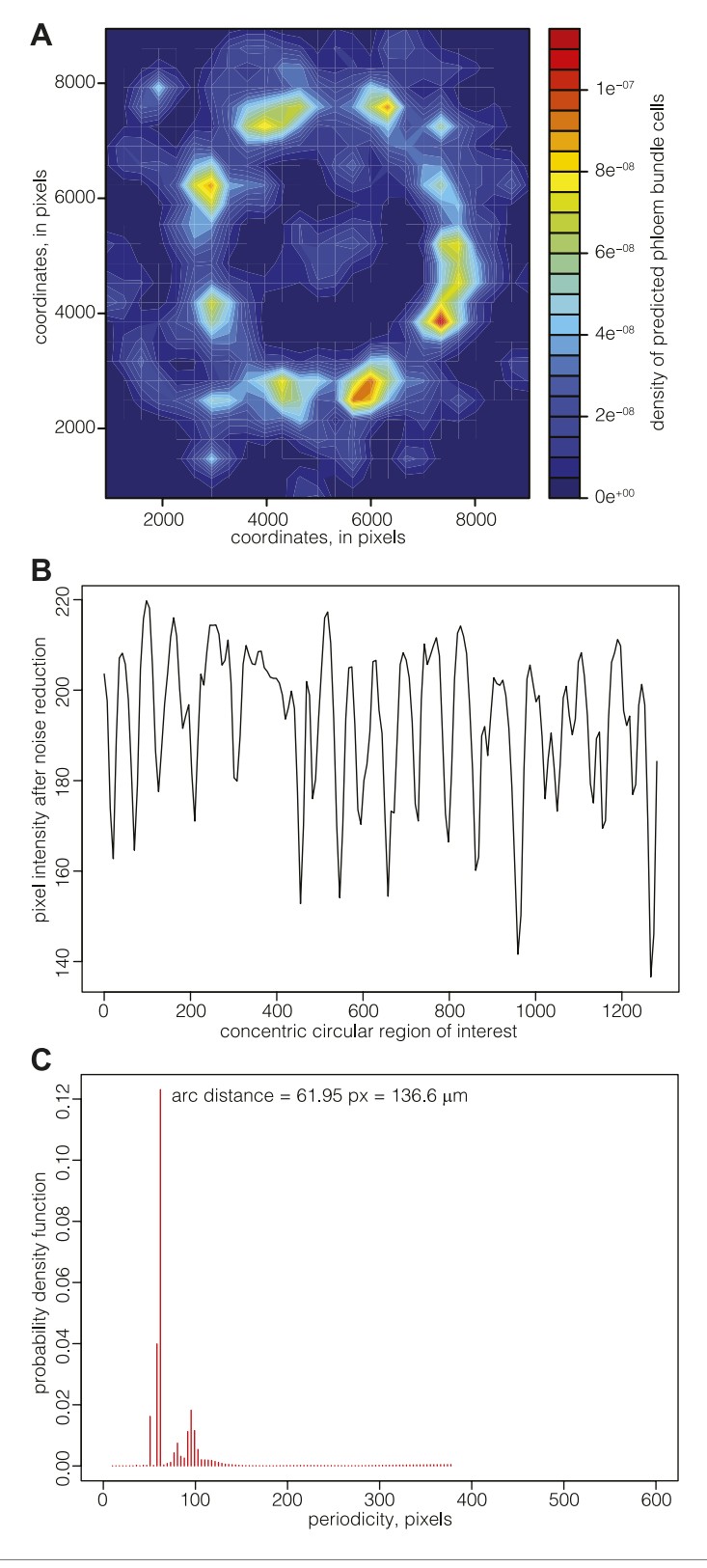

**Figure 6**. Mapping of phloem pole patterning. (**A**) Example of Gaussian kernel density estimate of the location of predicted phloem bundles cells in a 30 dag Col-0 section. High density represents phloem poles. (**B**) Example of an analysis of emerging phloem pole position in a 30 dag Col-0 section. The plot represents a pixel intensity map after

*Figure 6. Continued on next page*

*Figure 6. Continued*

noise reduction along a circular region of interest across the emerging phloem poles. Intensity peaks are due to GUS staining conferred to phloem bundles by an *APL::GUS* reporter construct. (**C**) Probability density function of the data shown in (**B**) obtained from an automated Bayesian model. The dominant single peak indicates a constant arc distance of ca. 62 pixel between the phloem poles.

As a secondary growth system, we chose the Arabidopsis hypocotyl, which has been shown previously to pose various advantages as opposed to Arabidopsis stems, most notably the uncoupling of elongation growth and secondary growth (*Sibout et al., 2008*). While a high-throughput approach was necessary to obtain statistically solid data, high-resolution imaging was required for reliable cellular level analyses. A novel type of global approach, that is, quantitative histology combined with machine learning, was the only realistic option to achieve both goals.

## Quantitative histology, an automated and machine learning-based approach

The principal problems that we faced were the large range of cell sizes as well as the large number of objects within the hypocotyl radius. This required ultra high-resolution imaging of our cross sections as well as an automated segmentation procedure that would not require any seeding. The solution was the assembly of cross sections from tiled, partial high-resolution images and their segmentation through an automated pipeline that relied on a watershed algorithm. This pipeline achieved very good accuracy in object detection, but was still CPU intensive. In part, this could be off set by binarization of the images using an adaptive Gaussian filter, which greatly accelerated the segmentation procedure. We could compensate an associated decrease in segmentation quality (because watershed segmentation is more accurate on gray scale images) by effectuating morphological operations on the binarized images, thereby keeping segmentation accuracy high while automating the task. Extending our approach beyond simple cell counting to cell type recognition intrinsically hinged on supervised classification. To this end, we used the Support Vector Machine (SVM) method, because it had already proven its efficiency in a broad range of life science applications (*Noble, 2006*). Average prediction accuracy based on this method was generally high, however for some cell type categories it was more variable at times. This was due to the nature of the classifiers, which were chosen to optimize for overall accuracy including all cell type categories. Implementation of our quality control tool alleviated this effect, however it is noteworthy that even more accurate classifiers can be identified for analyses that focus on a given cell type or a given time point, extending the range of potential applications of our pipeline.

## Morphology-based classification of plant cells

The use of shape characteristics for cell classification was pioneered by Olson et al., who classified mammalian culture cells into three groups using hierarchical cluster analysis and nearest neighbor analysis (*Olson et al., 1980*). Recent improvements in this area largely benefit from SVM algorithm development, which can take multiple features into account. For instance, a recent study identified factors involved in the transition between cell shapes using automated phenotyping of human cell cultures that took advantage of fluorescent staining for DNA, tubulin and actin on top of cell morphology (*Fuchs et al., 2010*). Conceptually similar, another study exploited cell shape in combination with fluorescent characteristics upon nuclear and cytoskeleton staining in Drosophila (*Yin et al., 2013*). However, classification based solely on cell morphology has also been applied to human cells (*Theriault et al., 2012*). Whereas all of these studies investigated isolated cells in culture, we had to apply morphology-based classification to cells that were embedded in their tissue and in a developmental context. While this complicated the analysis, it also offered the opportunity to assign spatial coordinates to the cells, which could be integrated on top of characteristics of cell geometry to build our classifiers. Average true prediction accuracy in the cited studies was in the range of 83–90%, as compared to 88% in our study. Notably however, our cell type assignment precision was greatly increased by our post-machine learning quality control pipeline, which enabled us to fix the principal classes with lower accuracy, due to frequent SVM confusion between xylem vessels and phloem parenchyma cells. Thereby, we successfully classified up to five cell type categories in a time course experiment where the number of cells ranged from a few hundred to several thousand. The factors that limited our approach were to some degree related to the properties of plant cells, notably that they are encapsulated by

rigid cell walls that resist the internal turgor pressure. Their cellular geometry is therefore not only shaped by the material properties of the walls, but also by the permanent force of turgor pressure, manifesting in the reduced variation of cell shape in plants as compared to animals (*Theriault et al., 2012*). To some degree, this uniformity in cell shape hampered the identification of certain cell states by our machine learning approach, for instance the direct identification of dividing cells. Similarly, certain cell types were ticklish to distinguish by their morphology only. For instance, we were not able to separate phloem companion cells from sieve elements or xylem parenchyma cells from xylem vessels across all time points, which therefore had to be grouped into combined categories. Adding tissue-related features, such as cell connectivity (i.e., the number of neighboring cells), and improving the segmentation algorithm such that cell wall thickness could be incorporated into the analyses might overcome these obstacles and greatly increase performance. Future efforts should go into this direction and could also boost the universal application of our approach. The latter should be possible for any tissue or organ from which cell outlines can be segmented after imaging and for which a reference point can be defined, for example (partial) sections from tree trunks or confocal images of root meristems.

## Vascular morphodynamics—a combination of molecular patterning and mechanical constraints?

For the subsequent cellular level analysis, the incline angle descriptor of a cell proved to be particularly valuable. Whereas no temporal changes were discernible for the cell area and the cell eccentricity features, the cell incline distribution varied over time, in a seemingly non-random fashion. Indeed, combination with spatial components (i.e., radial cell position in cross sections) revealed spatio-temporal rearrangement of inclines across a sequence of intertwined morphodynamic events. Our data indicate a gradual increase and arrangement of cambial cells, which together with orthoradial cellular organization of the surrounding tissues appeared to be a prerequisite for proper xylem development and relative xylem expansion around 20–25 dag. One possible explanation for this phenomenon could be tissue mechanics. The growing xylem area might exert a compression force on surrounding cambial and parenchymal cells, forcing them into tangential anisotropic cell elongation. How such mechanical stress is perceived and conveyed into cellular behavior is largely enigmatic and an emerging hot topic in plant biology, where first studies on shoot apical meristem formation have implicated katanins in the dynamic reorientation of microtubules perpendicular to stress direction (*Uyttewaal et al., 2012*).

Beyond possible mechanical constraints, molecular genetic patterning is clearly pivotal in vascular morphodynamics. For instance, polarity of the cambium to produce xylem to the inside and phloem to the outside is an inherent feature of secondary growth. A receptor-like kinase—peptide ligand pair is involved in this process and interacts with hormone signaling pathways (*Hirakawa et al., 2008*, *2010*; *Etchells et al., 2012*). Notably, the phenotypic penetrance of the respective mutants is background-dependent, with stronger effects in Ler than in Col-0 (*Etchells et al., 2013*). It would be interesting to investigate whether this could result from an interaction with the earlier cessation of phloem production we observed in Ler.

## Differential secondary growth dynamics in Col-0 vs Ler

The early cessation of phloem production in Ler as compared to Col-0 does, however, not reflect an earlier termination of overall growth in Ler. Rather it appears that phloem production in Ler ceases before xylem production and contributes to the divergent growth dynamics in the two genotypes. The severely reduced overall cell production in Ler as compared to Col-0 can be mainly attributed to reduced phloem and cambium cell number, and is responsible for the higher relative proportion of xylem area that had been reported earlier (*Ragni et al., 2011*). Interestingly, the nearly 50% reduction in overall cell number does not mean that growth is uniformly slower in Ler. Rather, initial secondary growth appears to be particularly slow in Ler as indicated by more than threefold difference in cell number at 15 dag. This is followed by an acceleration of cell production that surpasses Col-0 in relative terms between 15 dag and 25 dag, before dropping to Col-0 levels between 25 dag and 35 dag. This pattern is also evident from the principal component analysis, in which both Col-0 and Ler reach overall similar end points. Thus, our analysis along a series of time points has revealed highly divergent secondary growth dynamics in the genotypes that would not have been evident from a comparison of end points.

## Morphometric evidence for a phloem patterning mechanism

Beyond the cellular dimensions, our quantitative histology approach also allowed us to conduct follow up analyses to reveal developmental patterns that were not evident from simple visual inspection. For instance, we found a constant arc interspace distance for phloem pole formation along the developmental time series with a concomitant decrease in the interspace angle due to the overall secondary growth. A reaction-diffusion model with a growing boundary (i.e., representing the expanding xylem area) would be consistent with these results. Local production of the above-mentioned mobile ligand and activation of its receptor at a distance are potential candidates for such a mechanism. Alternatively, patterning cues from apical sources might direct phloem pole formation, for instance to coordinate it with phyllotaxy. Application of our quantitative histology approach to complementary stem sections could present one way to explore these possibilities.

## Materials and methods

### Plant material, sectioning and image acquisition

The *Arabidopsis thaliana* Col-0, Ler or *APL::GUS* (*Bonke et al., 2003*) lines were grown in soil, in a 16 hr light–8 hr dark cycle mimicking long day conditions under white light of ca. 150 µE intensity. After harvest, hypocotyls were immediately fixed and embedded in Technovit plastic resin before toluidine blue staining as described (*Sibout et al., 2008*; *Ragni et al., 2011*). Sections of 3-µm thickness were then obtained using a Leica RM2255 microtome and were subsequently imaged on a Zeiss LSM 710 confocal microscope in transmitted light mode at 40x magnification using the automated tiling function. Hypocotyls from *APL::GUS* plants were subjected to GUS staining before fixation, embedding and sectioning as described (*Sibout et al., 2008*; *Ragni et al., 2011*) and imaged using a Leica DM 5500 microscope.

### Quantitative histology

To extract information content of the sections at cellular resolution, we developed an automated image analysis pipeline. The pipeline was written in python with calls to R scripts and ImageJ macros. In brief, images were first pre-processed automatically (i.e., gamma correction, contrast, and brightness adjustment) before their binarization. A series of morphological operations (two times an erosion operation followed by a dilatation operation) were applied with the aim to discard noisy pixels and regularize the cell boundaries. These steps were achieved using to the EBImage R package (*Pau et al., 2010*). A variant watershed algorithm with automatic seeding (http://bigwww.epfl.ch/sage/soft/watershed) was used to identify the cell boundaries. Each cell was then characterized by a vector composed of 16 components that comprised 10 geometrical and 6 positional features (*Supplementary file 1A*) and was classified into one of the 5 cell-type classes (*Supplementary file 1B*). One classifier was built per genotype and per time point (*Supplementary file 1C*) using the *C*-classification with a radial basis function (RBF) kernel of the support vector machine (*Cortes and Vapnik, 1995*) provided by the e1071 package, the R interface for the libsvm library (*Chang and Lin, 2001*). The training set for the machine learning comprised 3144 manually labeled cells across 20 sections that covered all time points and genotypes. The optimal parameters, the selected features and the classifier accuracies are given in *Supplementary file 1D*.

### Phenoprints and comparison

To compare secondary growth progression in the two genotypes, we described each developmental stage in a 'phenoprint' that represents a vector combined of 8 numerical values (*Figure 2B*). For principal component analysis (PCA), each observable was scaled with respect to the maximum value to obtain a unit range across variables (*Supplementary file 4*). We performed a PCA analysis by computing the eigenvalues and eigenvectors for the correlation matrix. The resulting two first principal components were displayed with a bi-plot representation. The rotation angle between the vector variables represents the correlation between two phenoprints (>90° meaning no correlation). This method allowed direct quantitative comparison of the phenotypic variability of our samples.

### Phloem pole pattern analysis

To automatically map phloem pole positions in sections obtained from *APL::GUS* plants, a circular region of interest (ROI) across the newly generated phloem bundles that was concentric with the section center was defined and GUS staining intensity was measured along the ROI using ImageJ software. For each image, the period between phloem poles was detected using an automated

Bayesian model (*Granqvist et al., 2012*), corresponding to the most likely occurring arc interspace distance between two phloem poles.

## Acknowledgements

We would like to thank the Swiss Institute of Bioinformatics Vital-IT platform for support in computational infrastructure, Dr A Rodriguez-Villalon for the seedling photo, F Misceo for GUS-stained sections and Prof Ted Farmer for coining the term 'Quantitative Histology'.

## Additional information

### Competing interests

CSH: Reviewing Editor, *eLife*. The other authors declare that no competing interests exist.

### Funding

| Funder | Author |
|---|---|
| SystemsX | Ioannis Xenarios, Christian S Hardtke |
| EMBO longterm post-doctoral fellowships | Kaisa Nieminen, Laura Ragni |
| Marie Heim-Voegtlin | Laura Ragni |
| University of Lausanne | Martial Sankar, Christian S Hardtke |

The funders had no role in study design, data collection and interpretation, or the decision to submit the work for publication.

### Author contributions

MS, KN, LR, Conception and design, Acquisition of data, Analysis and interpretation of data, Drafting or revising the article; IX, Conception and design; CSH, Conception and design, Analysis and interpretation of data, Drafting or revising the article

## Additional files

### Supplementary files

• Supplementary file 1. (**A**) An explanation of the extracted parameters that describe the cellular features. (**B**) Summary information of the hand-labeled training set for supervised machine learning. (**C**) Definition of the classifiers selected for analysis. (**D**) Summary of the classifier parameters for supervised machine learning. (**E**) Overview of the cell type classes recognized by the supervised machine learning approach and their assignment codes used in Data Files 3 and 4.

• Supplementary file 2. Quality control files for the Col-0 sections.

• Supplementary file 3. Quality control files for the Ler sections.

• Supplementary file 4. The normalized values of the phenoprints (*Figure 2B*) used for PCA.

### Major dataset

The following dataset was generated:

| Author(s) | Year | Dataset title | Dataset ID and/or URL | Database, license, and accessibility information |
|---|---|---|---|---|
| Sankar M, Nieminen K, Ragni L, Xenarios I, Hardtke CS | 2014 | Data from: Automated quantitative histology reveals vascular morphodynamics during Arabidopsis hypocotyl secondary growth | http://dx.doi.org/10.5061/dryad.b835k | Available at the Dryad Digital Repository. |

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
