## [Decision Letter]

Thank you for sending your work entitled “Automated quantitative histology reveals vascular morphodynamics during Arabidopsis hypocotyl secondary growth” for consideration at *eLife*. Your article has been favorably evaluated by a Senior editor, Detlef Weigel, and 3 reviewers, one of whom served as the guest Reviewing editor for this article.

All three reviewers agree that this study describes a robust tool that represents a broadly useful addition to existing methods.

Nevertheless a number of issues have been identified that need to be addressed before the article is acceptable for publication:

1) While the method is well adapted to the biological system analysed here, it would be important to have an idea of the applicability to other organs and/or species. Would, for example, the pipeline function as well in the case of cambium formation in poplar or root cell differentiation in maize? Since this article is mainly technically oriented and the data presented here only provide limited further biological insight, it will be important to underline and fully explain the methodological significance of the work described here. While this does not necessarily imply that extra experiments are required, it should be made clear what the wider applications are. This would largely compensate for the lack of clear conclusions on the biological system.

2) The cell type detection has an accuracy of 88%, which seems relatively low. The key criteria used by the classifier to distinguish between the cell types are not very clear. If for example the main observable used to make the distinction between the cell types turns out to be cell size, then having a 12% miss-classification could have quite some effect on the conclusions drawn in the paper.

3) Two remarks concern the PCA analysis: 

- One of the reviewers performed a PCA on the data from Table 2 (using R software ade4 package) and could not reproduce the results presented in Figure 3. The authors should clarify what data they used for this PCA. If the PCA was indeed done on Table 2B, they should double check this part of their analysis. The reviewer suggested also to include intermediate steps of the PCA (correlation matrix, eigenvectors) as supplementary material.

- There is no discussion in the paper as to how the different observables contribute to the first principle component, which represents almost 94% of the variation. What is actually explaining almost all of this variation? Such a discussion would make it much more insightful what is actually changing over time and what makes Col-0 different from Ler.

4) In the part on “Visualization of vascular morphodynamics through combined plots of cell size and incline angle” there seems to be an issue with the incline angle: what happens when a cell is round? One would expect a highly randomized distribution of incline angles in that case. Please indicate how this problem was addressed.

5) Several points concern the more biological implications of the work described: 

- The paper contains a long description regarding the differences between the two genetic backgrounds in terms of total cross-sectional area, size variations and so forth, but no context is given how this information can be useful for understanding *Arabidopsis* development.

- In principle it should be possible to derive the relative contribution of cell expansion and cell proliferation from the data (see for example the Supporting Online Material of Bosveld et al., Science 2012). This would show how without having the availability of explicit time series, the cell dynamics underlying secondary growth can still be derived through statistical measures. Although such an analysis might be beyond the scope of this paper, it would help the paper to go beyond methodology.

- In general, the papers suffers from giving many precise measurements without inserting them in a proper context, such that it becomes unclear why these specifics are insightful and important for understanding plant development.

You might try to be clearer about these biological implications in both the Results and the Discussion.

---

## [Author Response]

*1) While the method is well adapted to the biological system analysed here, it would be important to have an idea of the applicability to other organs and/or species. Would, for example, the pipeline function as well in the case of cambium formation in poplar or root cell differentiation in maize? Since this article is mainly technically oriented and the data presented here only provide limited further biological insight, it will be important to underline and fully explain the methodological significance of the work described here. While this does not necessarily imply that extra experiments are required, it should be made clear what the wider applications are. This would largely compensate for the lack of clear conclusions on the biological system*.

It is true that our study is above all a proof of principal for the applicability of our approach, but it should work in any context where cell outlines can be reliably segmented and a reference point in the tissue can be defined. We have now added a few sentences in the Discussion to clarify this point:

“The latter [approach] should be possible for any tissue or organ from which cell outlines can be segmented after imaging and for which a reference point can be defined, e.g., (partial) sections from tree trunks or confocal images of root meristems.”

We have also looked into running our pipeline on alternative templates; however we could not obtain a sufficient number of consistently imaged high quality samples of a given tissue to perform such an analysis. Part of the problem is that already a reasonable amount of images are needed for the training set, before an automated run can even be launched.

*2) The cell type detection has an accuracy of 88%, which seems relatively low. The key criteria used by the classifier to distinguish between the cell types are not very clear. If for example the main observable used to make the distinction between the cell types turns out to be cell size, then having a 12% miss-classification could have quite some effect on the conclusions drawn in the paper*.

First, let us comment on the prediction accuracy. It is true that 88% might seem relatively low; however it compares favorably with other studies, which are typically in the same range or below, despite an at times reduced complexity. We have now added some more sentences to highlight this issue in the discussion (the newly cited studies have been added to the references):

“Conceptually similar, another study exploited cell shape in combination with fluorescent characteristics upon nuclear and cytoskeleton staining in *Drosophila* (27). However, classification based solely on cell morphology has also been applied to human cells (25). Whereas all of these studies investigated isolated cells in culture, we had to apply morphology-based classification to cells that were embedded in their tissue and in a developmental context. While this complicated the analysis, it also offered the opportunity to assign spatial coordinates to the cells, which could be integrated on top of characteristics of cell geometry to build our classifiers. Average true prediction accuracy in the cited studies was in the range of 83–90%, as compared to 88% in our study. Notably however, our cell type assignment precision was greatly increased by our post- machine learning quality control pipeline, which enabled us to fix the principal classes with lower accuracy, due to frequent SVM confusion between xylem vessels and phloem parenchyma cells.”

Please also pay attention to the last sentence above. That is, it is important to note that the quality control pipeline that we have implemented to correct mis-assignments (see Results section *Automated quality control and refinement of cell type recognition*) has greatly improved our final cell type classification reliability, which is thus more accurate than the initial performance of the machine learning. We hope that the modification of the Discussion clarifies this now.

Second, regarding the main observables, the reviewers highlight the main disadvantage of SVM regarding other machine learning techniques: whereas SVM can perform non-linear classification and “easily” handle multi-class problems, it does not permit to see which criteria have the most influence in the decision boundary. This is contrary to other methods such as decision tree or regression, which have a better interpretability (and perform well on binary problems but do not allow non-linear separation). For better documentation, we have now included an illustration of classifier selection by the V-fold cross validation method (Supplementary file 3), and we have added a new table (Supplementary file 4) that recapitulates the different qualifiers and the features they combine. This table shows that optimal classifiers varied between time points and genotypes, with no prevalent observable, such as cell size, dominating throughout.

*3) Two remarks concern the PCA analysis*:

*- One of the reviewers performed a PCA on the data from Table 2 (using R software ade4 package) and could not reproduce the results presented in*
Figure 3*. The authors should clarify what data they used for this PCA. If the PCA was indeed done on Table 2B, they should double check this part of their analysis. The reviewer suggested also to include intermediate steps of the PCA (correlation matrix, eigenvectors) as supplementary material*.

*- There is no discussion in the paper as to how the different observables contribute to the first principle component, which represents almost 94% of the variation. What is actually explaining almost all of this variation? Such a discussion would make it much more insightful what is actually changing over time and what makes Col-0 different from Ler*.

We thank the reviewers for bringing this to our attention. First, let us apologize for a mistake in some of the values for Ler in the phenoprint table (Figure 2). This was due to confusion by the corresponding author during figure assembly (average vs median values) and has been corrected now (the new values are close to the old ones). Moreover, the phenoprint table was indicative. The actual PCA input differed by the fact that we used the average radius of the section instead of the section surface area, which causes the difference in the PCA results obtained by the reviewer, independently of the software package used. To avoid any further confusion, we reconsidered the PCA by taking as input the exact same values displayed in Figure 2 (bimodal p value is used without the -log10 transformation). Since PCA is sensitive to scale, we corrected each descriptor value by its maximum value to obtain normalized unit range for all data. This input table is provided now as Supplementary file 13 as indicated in the text:

“The phenoprints consisted of a set of eight multi-parametric descriptors, which was informative for the normalized values (Supplementary file 13) that were used to perform a principal component analysis (Figure 3).”

Accordingly, we have revised Figure 3 and as requested now also display the observables and eigenvalues. This effectively refines our interpretation of temporal changes in Col-0 and Ler. The text in the manuscript has been modified accordingly and now points out what explains most of the variation:

“The computed correlation matrix was projected into a two-dimensional coordinate system, with the first two principal components explaining 76 % of the variation. The first component opposed the larger phenoprint stages (30 to 35 dag in both genotypes) with the smallest (Ler 15d), with proportionally less cambium in the older stages. The second component associated variables of large phloem proportion and inexistent or low fiber content (Col-0 15 dag, Ler 25 dag, Col-0 20 dag, Col-0 25 dag). The analysis also revealed larger angle spans for Ler as compared to Col-0 above all between 15 dag and 25 dag, suggesting substantial morphological changes during the early stages. At later time points, the two genotypes increasingly clustered together, indicating an initially slower development in Ler that however eventually caught up with Col-0.”

The main conclusion forom the PCA remains the same:

“Overall, the phenoprint clustering suggests a conserved sequence of development from one distinct morphological pattern to another, albeit with a different temporal progression in Col-0 versus Ler.”

*4) In the part on “Visualization of vascular morphodynamics through combined plots of cell size and incline angle” there seems to be an issue with the incline angle: what happens when a cell is round? One would expect a highly randomized distribution of incline angles in that case. Please indicate how this problem was addressed*.

This is a good point and was indeed one of our initial concerns. It might in part be responsible for more random incline angles at early stages. However, in practice, it turned out that round cells were very rare, as indicated by our eccentricity parameter (minor axis divided by major axis length), which was (mostly much more) smaller than 0.95 in typically more than 99% of cells for a given section.

*5) Several points concern the more biological implications of the work described*:

*- The paper contains a long description regarding the differences between the two genetic backgrounds in terms of total cross-sectional area, size variations and so forth, but no context is given how this information can be useful for understanding* Arabidopsis *development*.

*- In principle it should be possible to derive the relative contribution of cell expansion and cell proliferation from the data (see for example the Supporting Online Material of Bosveld et al., Science 2012). This would show how without having the availability of explicit time series, the cell dynamics underlying secondary growth can still be derived through statistical measures. Although such an analysis might be beyond the scope of this paper, it would help the paper to go beyond methodology*.

*- In general, the papers suffers from giving many precise measurements without inserting them in a proper context, such that it becomes unclear why these specifics are insightful and important for understanding plant development*.

*You might try to be clearer about these biological implications in both the Results and the Discussion*.

First, let us comment on the derivation of cell dynamics underlying secondary growth through statistical measures. Using high-resolution live imaging, Bosveld et al. performed a fine and precise quantitative analysis of cellular features and were able to assess the contribution of the cells’ shape changes and rearrangements to tissue morphogenesis. To do so, they used an original method based on a formalism applied in foam dynamics and they used a Fast Fourier Transform method to register (i.e., align) time-lapse movies of several individuals, thus obtaining robust statistics. In our case, the coarse timing prevents such an elegant analysis; rather we need a computational model of tissue dynamics to infer the contribution of cell expansion and cell proliferation on vascular tissue morphogenesis. We are actively working on this, but have not yet succeeded in creating a model, which will still take considerable time and which we believe is out of the scope of this study.

Regarding the biological implications of our results, it is true that we have been rather concise on this point. We have now elaborated on our findings, such as the equidistant phloem pole patterning or the masking of growth dynamics by the sole analysis of end points, and we have added a paragraph to the Discussion that highlights the main findings with regards to divergent secondary growth dynamics between Col-0 and Ler:

“Differential secondary growth dynamics in Col-0 versus Ler. The early cessation of phloem production in Ler as compared to Col-0 does, however, not reflect an earlier termination of overall growth in Ler. Rather it appears that phloem production in Ler ceases before xylem production and contributes to the divergent growth dynamics in the two genotypes. The severely reduced overall cell production in Ler as compared to Col-0 can be mainly attributed to reduced phloem and cambium cell number, and is responsible for the higher relative proportion of xylem area that had been reported earlier (21). Interestingly, the nearly 50 % reduction in overall cell number does not mean that growth is uniformly slower in Ler. Rather, initial secondary growth appears to be particularly slow in Ler as indicated by the more than three-fold difference in cell number at 15 dag. This is followed by an acceleration of cell production that surpasses Col-0 in relative terms between 15 dag and 25 dag, before dropping to Col-0 levels between 25 dag and 35 dag. This pattern is also evident from the principal component analysis, in which both Col-0 and Ler reach overall similar end points. Thus, our analysis along a series of time points has revealed highly divergent secondary growth dynamics in the genotypes that would not have been evident from a comparison of end points.”